# Molecular movie of ultrafast coherent rotational dynamics of OCS

Evangelos T. Karamatskos [1,2], Sebastian Raabe[3], Terry Mullins[1], Andrea Trabattoni[1,2], Philipp Stammer[3], Gildas Goldsztejn[3], Rasmus R. Johansen[4], Karol Długołecki[1], Henrik Stapelfeldt [4], Marc J.J. Vrakking[3], Sebastian Trippel[1,5], Arnaud Rouzée[3] & Jochen Küpper [1,2,5]

Recording molecular movies on ultrafast timescales has been a longstanding goal for unravelling detailed information about molecular dynamics. Here we present the direct experimental recording of very-high-resolution and -fidelity molecular movies over more than one-and-a-half periods of the laser-induced rotational dynamics of carbonylsulfide (OCS) molecules. Utilising the combination of single quantum-state selection and an optimised two-pulse sequence to create a tailored rotational wavepacket, an unprecedented degree of field-free alignment, $\langle \cos^2 \theta_{2D} \rangle = 0.96$ ($\langle \cos^2 \theta \rangle = 0.94$) is achieved, exceeding the theoretical limit for single-pulse alignment. The very rich experimentally observed quantum dynamics is fully recovered by the angular probability distribution obtained from solutions of the time-dependent Schrödinger equation with parameters refined against the experiment. The populations and phases of rotational states in the retrieved time-dependent three-dimensional wavepacket rationalises the observed very high degree of alignment.

[1] Center for Free-Electron Laser Science, Deutsches Elektronen-Synchrotron DESY, Notkestraße 85, 22607 Hamburg, Germany. [2] Department of Physics, Universität Hamburg, Luruper Chaussee 149, 22761 Hamburg, Germany. [3] Max Born Institute, Max-Born-Straße 2a, 12489 Berlin, Germany. [4] Department of Chemistry, Aarhus University, Langelandsgade 140, 8000 Aarhus C, Denmark. [5] The Hamburg Center for Ultrafast Imaging, Universität Hamburg, Luruper Chaussee 149, 22761 Hamburg, Germany. Correspondence and requests for materials should be addressed to A.R. (email: arnaud.rouzee@mbi.de) or to J.K. (email: jochen.kuepper@cfel.de)

The filming of nuclear motion during molecular dynamics at relevant timescales, dubbed the "molecular movie", has been a longstanding dream in the molecular sciences[1,2]. Recent experimental advances with X-ray-free-electron lasers and ultrashort-pulse electron guns have provided first glimpses of intrinsic molecular structures[3–5] and dynamics[2,6,7]. However, despite the spectacular progress, the fidelity of the recorded movies, in comparison to the investigated dynamics, was limited so far. Especially for high-precision studies of small molecules, typically only distances between a few atoms were determined[4,5,7].

Rotational quantum dynamics of isolated molecules provides an interesting and important testbed that provides and requires direct access to angular coordinates. Furthermore, different from most molecular processes, it can be practically exactly described by current numerical methods, even for complex molecules. Rotational wavepackets were produced through the interaction of the molecule with short laser pulses[8–10], which couple different rotational states through stimulated Raman transitions. The resulting dynamics were observed, for instance, by time-delayed Coulomb-explosion ion imaging[9,11,12], photoelectron imaging[13] or ultrafast electron diffraction[14]. The rotational wavepackets were exploited to connect the molecular and laboratory frames through strong-field alignment[9,10] and mixed-field orientation[15,16] as well as for the determination of molecular-structure information in rotational-coherence spectroscopy[17,18]. Coherent rotational wavepacket manipulation using multiple pulses[19] or appropriate turn-on and -off timing[20] allowed enhanced or diminished rephasing, and it was suggested as a realisation of quantum computing[19]. Furthermore, methods for rotational-wavepacket reconstruction of linear molecules[21] and for benzene[22] were reported.

Here, we demonstrate the direct experimental high-resolution imaging of the time-dependent angular probability-density distribution of a rotational wavepacket and its complete characterisation in terms of the populations and phases of field-free rotor states. Utilising a state-selected molecular sample and an optimised two-laser-pulse sequence, see Supplementary Note 1, a broad phase-locked rotational wavepacket was created. Using mid-infra-red-laser strong-field ionisation and Coulomb-explosion ion imaging, an unprecedented degree of field-free alignment of $\langle\cos^2\theta_{2D}\rangle = 0.96$, or $\langle\cos^2\theta\rangle = 0.94$, was obtained at the full revivals, whereas in between a rich angular dynamics was observed with very high resolution, from which the complete wavepacket could be uniquely derived. While the dynamics has low dimensionality, the resulting—purely experimentally obtained—movie provides a most direct realisation of the envisioned molecular movie. We point out that the data also is a measurement of a complete quantum carpet[23].

## Results

**Experimental approach.** In order to achieve such a high degree of alignment, better than the theoretical maximum of $\langle\cos^2\theta\rangle = 0.92$ for single-pulse alignment[24,25], we performed a pump–probe experiment with ground-state-selected carbonylsulfide (OCS) molecules[26], with >80% purity, as a showcase. Two off-resonant near-IR pump pulses of 800 nm central wavelength, separated by 38.1(1) ps and with a pulse duration of 250 fs, that is much shorter than the rotational period of OCS of 82.2 ps, were used to create the rotational wavepacket. These pulses were linearly polarised parallel to the detector plane. The probe pulse with a central wavelength of 1.75 μm was polarised perpendicularly to the detector plane to minimise the effects of geometric alignment and ensures that the observed

degree of alignment was a lower boundary of the real value. The probe pulse multiply ionised the molecules, resulting in Coulomb explosion into ionic fragments. Two-dimensional (2D) ion-momentum distributions of $O^+$ fragments, which reflect the orientation of the molecules in space at the instance of ionisation, were recorded by a velocity map imaging (VMI) spectrometer[27] for different time delays between the alignment pulse sequence and the probe pulse. Further details of the experimental setup are presented in the "Methods" section.

**Experimental movie.** In Fig. 1, snapshots of the experimentally recorded molecular movie, that is 2D ion-momentum distributions, are shown for several probe times covering a whole rotational period. The phase of 0 and $2\pi$ correspond to $t = 38.57$ and 120.78 ps after the peak of the first alignment-laser pulse at $t = 0$, respectively. The simplest snapshot-images, reflecting an unprecedented degree of field-free alignment $\langle\cos^2\theta_{2D}\rangle = 0.96$, were obtained for the alignment revivals at phases of 0 and $2\pi$, which correspond to the prompt alignment and its revival regarding the second laser pulse. Here, the molecular axes are preferentially aligned along the alignment-laser polarisation. For the antialignment at a phase of $\pi$ the molecules are preferentially aligned in a plane perpendicular to the alignment-laser polarisation direction. Simple quadrupolar structures are observed at $\pi/2$ and $3\pi/2$. At intermediate times, at $\pi/3$ or $7\pi/12$, the images display rich angular structures, which could be observed due to the high angular experimental resolution of the recorded movie, which is 4° as derived in the Supplementary Note 4. This rich structure directly reflects the strongly quantum-state selected initial sample exploited in these measurements, whereas the structure would be largely lost in the summation of wavepackets from even a few initially populated states.

**Analysis of the rotational dynamics and the degree of alignment.** The dynamics was analysed as follows: through the interaction of the molecular ensemble with the alignment-laser pulses, a coherent wavepacket was created from each of the initially populated rotational states. These wavepackets were expressed as a coherent superposition of eigenfunctions of the field-free rotational Hamiltonian, that is

$$\Psi(\theta, \phi, t) = \sum_J a_J(t) Y_J^M(\theta, \phi), \tag{1}$$

with the time-dependent complex amplitudes $a_J(t)$, the spherical harmonics $Y_J^M(\theta, \phi)$, the quantum number of angular momentum $J$, and its projection $M$ onto the laboratory-fixed axis defined by the laser polarisation. We note that $M$ was conserved and thus no $\phi$ dependence existed. The angular distribution is defined as the sum of the squared moduli of all $\Psi(\theta, \phi, t)$ weighted by the initial-state populations.

The degree of alignment was extracted from the VMI images using the commonly utilised expectation value $\langle\cos^2\theta_{2D}\rangle$. The maximum value observed at the alignment revival reached 0.96, which, to the best of our knowledge, is the highest degree of field-free alignment achieved to date. Comparing the angular distributions at different delay times with the degree of alignment $\langle\cos^2\theta_{2D}\rangle$, see Supplementary Fig. 4, we observed the same degree of alignment for angular distributions that are in fact very different from each other. This highlights that much more information is contained in the angular distributions than in the commonly utilised expectation value[10]. Indeed $\langle\cos^2\theta_{2D}\rangle$, merely describes the leading term in an expansion of the angular distribution, for instance, in terms of Legendre polynomials, see

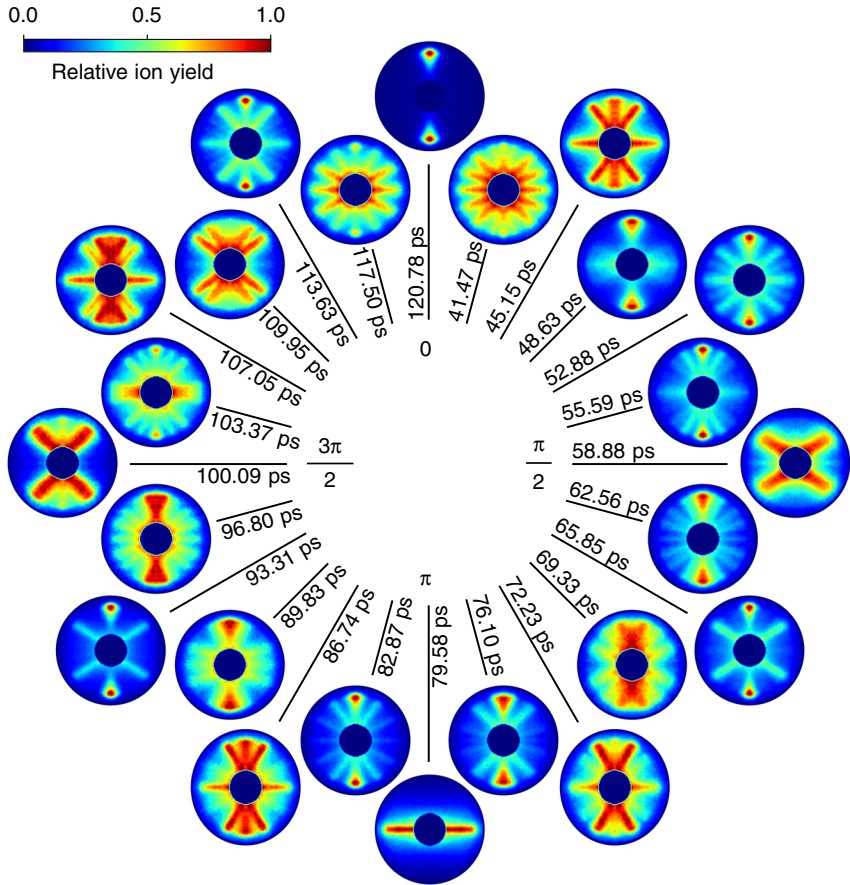

**Fig. 1** Rotational clock depicting the molecular movie of the observed quantum dynamics. Individual experimental VMI images of $O^+$ ion-momentum distributions depicting snapshots of the rotational wavepacket over one full period. The displayed data were recorded from the first (prompt) revival at 38.57 ps (0) to the first full revival at 120.78 ps ($2\pi$); the phase-evolution of $\pi/12$ between images corresponds to ~3.43 ps and the exact delay times of the individual images are specified. Full movies are available as Supplementary Movies 1 and 2

(1) in the Supplementary Note 2. In order to fully characterise the angular distribution a description in terms of a polynomial series is necessary that involves the same maximum order as the maximum angular momentum $J_{max}$ of the populated rotational eigenstates, which corresponds to, at most, $2J_{max}$ lobes in the momentum maps.

As the probe laser is polarised perpendicularly to the detector plane, the cylindrical symmetry as generated by the alignment-laser polarisation was broken and an Abel inversion to retrieve the 3D angular distribution directly from the experimental VMI images was not possible. In order to retrieve the complete 3D wavepacket, the time-dependent Schrödinger equation (TDSE) was solved for a rigid rotor coupled to a non-resonant ac electric field representing the two-laser pulses as well as the dc electric field of the VMI spectrometer. For a direct comparison with the experimental data the rotational wavepacket and thus the 3D angular distribution was constructed and, using a Monte-Carlo approach, projected onto a 2D screen using the radial distribution extracted from the experiment at the alignment revival at 120.78 ps. The relation between the 3D rotational wavepacket and the 2D projected density is graphically illustrated in Supplementary Fig. 2. The anisotropic angle-dependent ionisation efficiency for double ionisation, resulting in a two-body breakup into $O^+$ and $CS^+$ fragments, was taken into account by approximating it by the square of the measured single-electron ionisation rate. Non-axial recoil during the fragmentation process is expected to be negligible and was not included in the simulations.

**Fitting procedure and the computed molecular movie.** The initial-state distribution in the quantum-state selected OCS sample as well as the interaction volume with the alignment and probe lasers were not known a priori and used as fitting parameters. For each set of parameters the TDSE was solved and the 2D projection of the rotational density, averaged over the initial-state distribution and the interaction volume of the pump and probe lasers, was carried out. The aforementioned expansion in terms of Legendre polynomials was realised for the experimental and simulated angular distributions and the best fit was determined through least squares minimisation, see Supplementary Note 2. Taking into account the eight lowest even moments of the angular distribution allowed to precisely reproduce the experimental angular distribution. The results for the first four moments are shown in Fig. 2a; the full set is given in Supplementary Fig. 3 as well as the optimal fitting parameters in Supplementary Note 2. The overall agreement between experiment and theory is excellent for all moments. Before the onset of the second pulse, centred around $t = 38.1$ ps, the oscillatory structure in all moments is fairly slow compared to later times, which reflects the correspondingly small number of interfering states in the wavepacket before the second pulse, and the large number thereafter.

Theoretical images, computed for the best fit parameters, are shown in Fig. 2b; a full movie is provided as Supplementary Movie 1. The theoretical results are in excellent agreement with the measured ion-momentum angular distributions at all times,

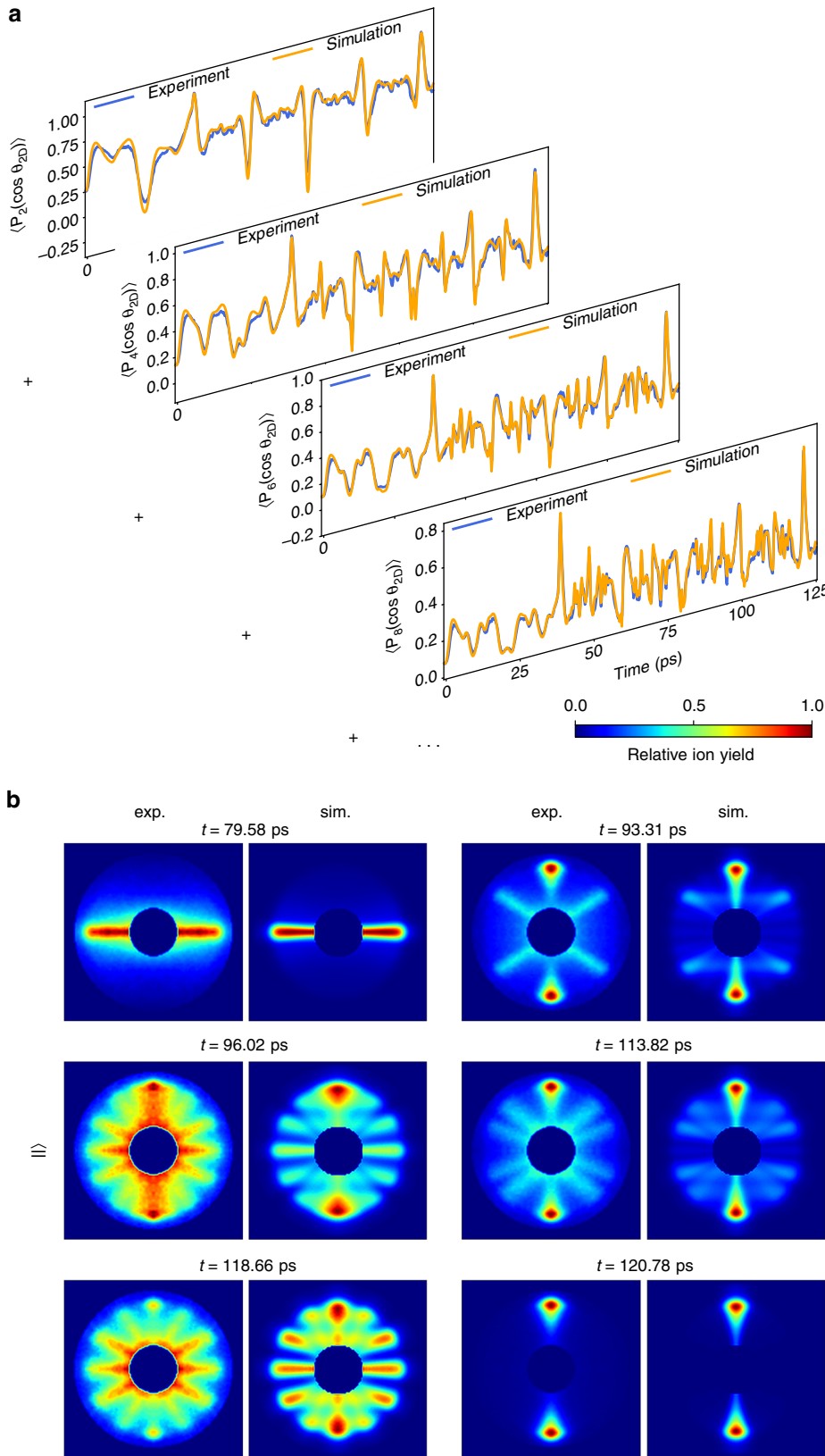

**Fig. 2** Decomposition of angular distributions into their moments. **a** Comparison of the decomposition of the experimental and theoretical angular distributions in terms of Legendre polynomials. **b** Simulated and experimental angular-distribution VMI images for selected times; the radial distributions in the simulations are extracted from the experimental distribution at 120.78 ps, see text for details

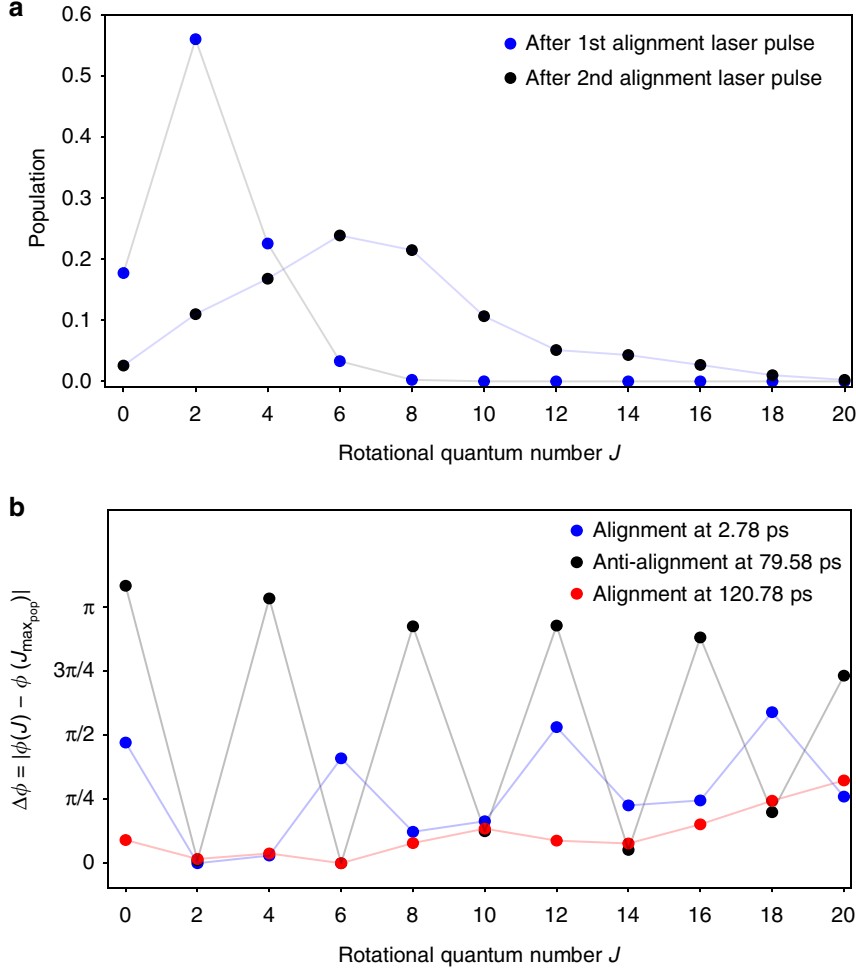

**Fig. 3** Populations and phase differences in the rotational wavepacket at alignment and antialignment times. **a** Rotational-state populations and **b** phase differences to the phase of the state with largest population, $J = 2$, $J = 6$, respectively, at the alignment revival following a single-pulse excitation, 2.78 ps (blue dots), and the two-pulse excitation, 120.78 ps (red dots) as well as for the antialignment at 79.58 ps (black dots, populations coincide with the red dots). Only states with even angular momentum are populated due to the Raman-transition selection rules $\Delta J = \pm 2$

see Supplementary Note 3, and prove that we were able to fully characterise the 3D rotational wavepacket with the amplitudes and phases of all rotational states included.

**Populations and phases in the wavepacket**. In Fig. 3a, the extracted rotational-state populations are shown for the wave-packet created from the rotational ground state after the first and the second alignment-laser pulse. It clearly shows that the rotational-state distribution is broader after the second pulse, reaching up to $J \geq 16$. This also matches the convergence of the Legendre-polynomial series, with eight even terms, derived from the fit to the data above. In Fig. 3b the corresponding phase differences for all populated states relative to the state with the largest population in the wavepacket are shown, where $\phi(J)$ is the phase of the complex coefficient $a_J$ in (1). Combining these populations and phases it became clear that the very high degree of alignment after the second alignment pulse arises from the combination of the broad distribution of rotational states, reaching large angular momenta, and the very strong and flat rephasing of all significantly populated states at the revival at 120.78 ps, Fig. 3b (red). Similarly, the antialignment at 79.58 ps occurs due to alternating phase differences of $\pi$ between adjacent populated rotational states, Fig. 3b (black).

## Discussion

We were able to record a high-resolution molecular movie of the ultrafast coherent rotational motion of impulsively aligned OCS molecules. State-selection and an optimised two-pulse sequence yielded an unprecedented degree of field-free alignment of $\langle \cos^2\theta_{2D} = 0.96 \rangle$, with a very narrow angular confinement of 13.4° FWHM, shown in Supplementary Note 5. Limiting the analysis to a determination of $\langle \cos^2\theta_{2D} \rangle$, as it is common in experiments on time-dependent alignment, did not allow to capture the rich rotational dynamics, while the use of a poly-nomial expansion up to an appropriate order did. We completely unravelled the rotational wavepacket, from which the complex coefficients and, hence, the full information about the rotational wavepacket under study was extracted. The 2D projection of the obtained rotational wavepacket allowed a direct comparison with the experimentally measured data.

Regarding the extension toward the investigation of chemical dynamics, we point out that strong-field-ionisation-induced Coulomb-explosion imaging can be used, for instance, to image the configuration of chiral molecules[28] or internal torsional dynamics[29]. Following the dynamics of such processes with the detail and quality presented here would directly yield a mole-cular movie of the chemical and, possibly, chirality dynamics[30]. Furthermore, the very high degree of field-free alignment

achieved here would be extremely useful for stereochemistry studies[31,32] as well as for molecular-frame imaging experiments[4,5,14,33–38].

## Methods

**Experimental setup**. A cold molecular beam was formed by supersonic expansion of a mixture of OCS (500 ppm) in helium, maintained at a backing pressure of 90 bar from a pulsed Even-Lavie valve[39] operated at 250 Hz. After passing two skimmers, the collimated molecular beam entered the Stark deflector. The beam was dispersed according to quantum state by a strong inhomogeneous electric field[26] with a nominal strength of ~200 kV/cm. Through a movable third skimmer, the molecular beam entered the spectrometer. Here, it was crossed at right angle by laser beams, where the height of the laser beams allowed to probe state-selected molecular ensembles, that is a practically pure rovibronic-ground-state sample of OCS[16,20,40].

The laser setup consisted of a commercial Ti:Sapphire laser system (KM labs) delivering pulses with 30 mJ pulse energy, 35 fs (full width at half maximum (FWHM)) pulse duration, and a central wavelength of 800 nm at a 1 kHz repetition rate. One part (20 mJ) of the laser output was used to pump a high-energy tunable optical parametric amplifier (HE-TOPAS, Light Conversion) to generate pulses with a central wavelengths of 1.75 μm, a pulse duration of 60 fs, and a pulse energy of ~1.5 mJ. Totally, 900 μJ of the remaining 800 nm laser output was used for the laser-induced alignment, that is the generation of the investigated rotational wavepackets. This beam was split into two parts with a 4:1 energy ratio using a Mach–Zehnder interferometer. A motorised delay stage in one beam path allowed for controlling the delay between the two pulses. This delay was optimised experimentally and maximum alignment was observed for $\tau_{exp} = 38.1 \pm 0.1$ ps, in perfect agreement with the theoretically predicted $\tau_{sim} = 38.2$ ps. The pulses were combined collinearly and passed through a 2 cm long $SF_{11}$ optical glass to stretch them to 250 fs pulse duration (FWHM). Then the alignment pulses were collinearly overlapped with the 1.75 μm mid-infra-red pulses using a dichroic mirror. All pulses were focused into the VMI spectrometer using a 25 cm focal-distance calcium fluoride lens.

At the centre of the VMI the state-selected molecular beam and the laser beams crossed at right angle. Following strong-field multiple ionisation of the molecules, the generated charged fragments were projected by the VMI onto a combined multichannel-plate phosphor-screen detector and read out by a charge-coupled device camera. The angular resolution of the imaging system is 4°, limited by the 1 megapixel camera, see Supplementary Note 4. 2D ion-momentum distributions of $O^+$ fragments were recorded as a function of the delay between the 800 nm pulses and the ionising 1.75 μm pulses in order to characterise the angular distribution of the molecules through Coulomb-explosion imaging. The polarisation of the 800 nm alignment pulses was parallel to the detector screen whereas that of the 1.75 μm ionising laser was perpendicular in order to avoid geometric-alignment effects in the angular distributions. For this geometry, unfortunately, it was not possible to retrieve 3D distributions from an inverse Abel transform. Totally, 651 images were recorded in steps of 193.4 fs, covering the time interval from −0.7 ps up to 125 ps, which is more than one-and-a-half times the rotational period of OCS of 82.2 ps.

## Data availability

All datasets generated in this study are available from the corresponding author on reasonable request. The original data are also available as the individual frames of the movie files in the Supplementary Movies 1 and 2 of this paper.

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

## Acknowledgements

This work has been supported by the Deutsche Forschungsgemeinschaft (DFG) through the priority programme "Quantum Dynamics in Tailored Intense Fields" (QUTIF, SPP1840, KU 1527/3, RO 4577/4) and by the Clusters of Excellence "Center for Ultrafast Imaging" (CUI, EXC 1074, ID 194651731) and "Advanced Imaging of Matter" (AIM, EXC 2056, ID 390715994) of the Deutsche Forschungsgemeinschaft (DFG) and by the European Research Council under the European Union's Seventh Framework Programme (FP7/2007-2013) through the Advanced Grant "DropletControl" (ERC-320459-Stapelfeldt) and the Consolidator Grant "COMOTION" (ERC-614507-Küpper).

## Author contributions

The project was conceived and coordinated by A.R. and J.K. The experiment was designed by K.D., J.K. and A.R.; set up by E.T.K., S.R., K.D., R.R.J. and A.R.; and performed by E.T.K., S.R., A.T., S.T., G.G., P.S. and A.R. The data analysis and numerical simulations were performed by E.T.K., the results from theory and experiment were analysed by E.T.K., T.M., A.T., S.T., A.R. and J.K. and discussed with H.S. and M.J.J.V. The paper was prepared by E.T.K., S.T. and J.K., and discussed by all authors.

## Additional information

**Competing interests:** The authors declare no competing interests.

