## [Peer Review File · Nature Communications]

Reviewers' comments:

Reviewer #1 (Remarks to the Author):

The manuscript reports 2D ion-imaging observation of coherent rotational motion which a pre-state selected molecular ensemble exhibits after the interaction with ultrashort intense laser pulses. As the authors pointed out, there are three major points which deserve the significance of the report, i.e., 1) the detailed experimental tracking of complex spatiotemporal propagation of molecular angular probability, 2) achievement of very high degree of molecular alignment, and 3) experimental retrieval of rotational wave packet (WP) created herein. So I would like to evaluate them in some details below.

1) Apparently, the highlight of the present manuscript is the successful recording of a series of 2D images that represent coherent rotational dynamics induced by ultrashort laser pulses. The experimental "movie" provided as a supplemental file is indeed enjoyable. The complex spatiotemporal propagation appeared in the movie is a clear manifestation of quantum interferences in the molecular rotational WP created therein. The observation was done with a conventional VMI setup, and the detailed and clear signature thus observed is mainly due to the two following efforts by the authors. First, the molecular ensemble to be examined was initially prepared as that restricted in the (almost) single lowest rotational level after the quantum-state selection with the electrostatic deflector, which was originally developed by the authors' research group. They have implemented such state selection of polar molecules to various researches already. Here, the restriction of the initial ensemble to the single level is essential to observe the quantum interference, which would be smeared out if the ensemble were composed of multiple rotational levels. The previous study by our research group (given in reference [11]) was along this direction, and the present report has reached to the limit. The second effort is the achievement of the excitation that makes the WP containing many rotational eigenstates with J ranging from 0 to ~ 15 . Such a high degree of state "mixing" inherently results in the rich and complex dynamics appeared in the "movie".

2) Since anisotropic molecular ensembles have been useful in various researches in the fields of molecular physics and quantum optics, achieving higher degree of molecular alignment is important. The present study reports the realization of field free molecular alignment exceeding the estimated limit for single-pulse excitation. It is really nice. The author adopted a double-pulse excitation scheme, which has been repeatedly examined and approved for improving molecular alignment, and found out the optimum condition from the model calculation based on the time-dependent Schrodinger equation (TDSE) calculation. Again, the restriction of the initial ensemble to the lowest single level is essential to achieve the highest alignment realized so far, and the development of the electrostatic deflector by the authors is crucial.

3) Retrieval of quantum states is also a big issue in molecular physics and quantum optics, and there have been many researches. For molecular rotational WPs, theoretical proposals and experimental demonstration have already been reported [A. S. Mouritzen and K. Molmer, *J. Chem. Phys.* 124, 244311 (2006); H. Hasegawa and Y. Ohshima, *Phys. Rev. Lett.* 101, 053002 (2008)], which the authors would cite. It is true that a series of 2D images for fragment ions from rotating molecules contains rich information on the rotational WP, but its extraction is not straightforward. The ionization/dissociation process is inherently anisotropic. When the linearly polarized light is used to induce Coulomb explosion, the ionization/dissociation probability is symmetric around the polarization direction but strongly depends on the angle between the polarization and the molecular axis. Furthermore, in the case of polyatomic molecules, like OCS, we have to take into account that the fragment recoil velocity may not direct along the molecular axis due to the ultrafast bending of the molecule during the ionization/dissociation. See, for example, J. H. Sanderson et al., *Phys. Rev. A* 65, 043403 (2002). Thus, when 2D ion imaging is adopted, as the authors did, where ions distributed in 3D are smashed onto the 2D detector, the observed images are given after convoluting the angular probability for the molecular direction (proportional to the absolute square of the WP) over

ionization/dissociation anisotropic probability as well as the angular distribution due to the bending dynamics. There is no description about this situation anywhere in the manuscript, and Eq. 2 in Supplemental Information implies that the authors may simply neglect other factors than that originates from WP. This is totally inappropriate. If the authors did make any consideration over other factors, it should be clearly described so that its validity and limitation are properly evaluated.

In addition to the above main points, there are several comments to be considered.

i) In FIG.3. $\varphi(J)$ is used without definition. Is it a total phase: $\varphi(J) = \omega_{J}t + \delta_{J}$?

ii) In Supplemental Information, p.4. It has to be clearly described: what are the floating parameters in the least squares fittings? Amplitudes and phases for each eigen functions (as has to be in conventional WP reconstruction, see Hasegawa and Ohshima), or others (e.g., laser pulse parameters)?

iii) P. 4, in the main text. What is the difference between the "2D-" and "3D-" reconstruction? What is the experimental input for the reconstruction (to fit the parameters) and what are the fitting parameters, for the both cases?

iv) In Supplemental Information, p.6. In an analysis (3D?), the initial state distribution with $J = 0$ to 2 is considered. How about the actual population for each state?

v) It should be pointed out, when the initial state distribution is spread among multiple levels, the resultant molecular ensemble after laser excitation cannot be represented as a single WP, but a mixed state. Even when the initial ensemble is in a single state, if the laser field variation is substantial in the probed area, the resultant ensemble is also in a mixed state. For both cases, a density matrix should be invoked to describe the state and any "WP reconstruction" cannot be adopted.

In conclusion, the manuscript contains very interesting finding pertinent to coherent rotational dynamics of an ultrashort-laser driven molecular ensemble. Still, there are several points, in particular mentioned in 3) above, to be seriously considered and substantially revised before publication.

Reviewer #2 (Remarks to the Author):

This manuscript presents a superb work by a team, which includes some of the pioneers of molecular alignment studies. The work utilizes all advanced alignment and detection techniques to achieve the highest degree of field-free alignment of OCS and to record a very detailed high-resolution images of time-dependent rotational nuclear wavepackets. I could not find any faults with either experiments or calculations. The paper is very well written, the results are clearly presented, the methods and approaches are well documented (in the paper and supplementary materials). The movie also looks cool. However, I cannot recommend this work for publication in Nature Communications. My opinion is that this work does not constitute a genuine breakthrough. While the authors seem to consider and do emphasize as their main achievement the high-resolution "molecular movie" of nuclear rotation, that process is generic for all linear molecules, very easily modelled numerically with an arbitrarily high accuracy and has been observed many times before, including by the authors of this manuscript themselves. There is very little actual benefit in the demonstrated high angular resolution in terms of gaining new insights or improving numerical models. Therefore, I am not so very impressed with the "movie" part. What I am impressed with (and what, I think, is the real main achievement of this work) is how it combines all the advanced techniques: rotational state selection, two-pulse impulsive alignment, Coulomb explosion imaging, VMI - to achieve near-perfect field-free alignment. Having said that, I will also note that all those techniques were already demonstrated separately some time ago, also by the authors themselves, and those papers are properly cited here too. Combining the established techniques is rather an incremental development, which while novel and important, in my view does not rise to the standard of Nature Communications. I appreciate, that this is a subjective judgement call based on comparing this work with the whole output of the journal, which is, perhaps, best done by the editors. Also, those dreaded "novelty and importance" are not so easily quantified, so

I have to go purely with my subjective opinion here. I feel that this manuscript would be more appropriate for Scientific Reports.

Reviewer #3 (Remarks to the Author):

In this manuscript the authors describe alignment experiments where a pair of nonresonant pulses generates a broad rotational wavepacket and Coulomb imaging is applied to probe the rotational distribution as a function of time. It is a beautiful paper and I recommend accepting it for publication in Nature Communications. I have, however, several comments that the authors may wish to consider.

1. The authors should explain the difference between $\langle \cos^2 \theta \rangle$ and $\langle \cos^2 \theta \rangle_{2d}$.
2. What is the accuracy of the measurement? Is the relative error small with respect to the difference between 0.96 and 0.92 (where the latter figure corresponds to a single initial eigenstate and one pulse)? I expect that the experimental accuracy is limited by the accuracy to which the intensity can be measured. With the error taken into account, 0.96 is not very different from the single pulse 0.92. This suggests that the good alignment demonstrated here is predominantly because of the single initial eigenstate. This agrees with early calculations.
3. Equation (1) and the details below are not needed. The dynamics of rotational wavepackets of simple, rigid rotor molecules have been computed before and the method is known. The authors could provide a reference to a theory paper or review where this is more fully explained and use the space to expand on the results.
4. P. 2, paragraph 2, 6 lines from the end: Why do the authors use Chebyshev polynomials? Please see Phys.Rev.Lett. 89 233002 (2002), from which it is evident that the correct expansion polynomials are the Legendre polynomials (or the Legendre functions in the more general case). The cross section could be expanded in terms of other polynomials but this does not have physical origin. Expansion in terms of Legendre polynomials yields the moments of the rotational distribution as the expansion coefficients. The same reference makes the argument made also here that the complete rotational density contains much more information (all moments) than the conventional $\langle \cos^2 \theta \rangle$.
5. Figure 2b: The authors argue several times that the agreement of the calculations with the measurements is excellent, but in reality, considering that the calculation used radial distributions extracted from the experiments, one could expect much better agreement of the images. Why is the resemblance poor?
6. The experimental images are not a measurement of the square magnitude of the wavepacket (as the manuscript argues) because they include the complex and system-dependent dynamics of ionization and fragmentation. Of course, the rotational distribution has a role in determining the images, basically because the measured angular distribution is tied to the rotational wavepacket angular distribution through angular momentum selection rules. Please see Phys.Rev. A 87, 023411 (2013) for explanation how the rotational wavepacket angular density relates to the observed image.

Remarks by referee 1

The manuscript reports 2D ion-imaging observation of coherent rotational motion which a pre-state selected molecular ensemble exhibits after the interaction with ultrashort intense laser pulses. As the authors pointed out, there are three major points which deserve the significance of the report, i.e., 1) the detailed experimental tracking of complex spatiotemporal propagation of molecular angular probability, 2) achievement of very high degree of molecular alignment, and 3) experimental retrieval of rotational wave packet (WP) created herein.

We thank the referee for his very positive evaluation. We agree with his summary of the most relevant parts of the presented work.

So I would like to evaluate them in some details below.

1) Apparently, the highlight of the present manuscript is the successful recording of a series of 2D images that represent coherent rotational dynamics induced by ultrashort laser pulses. The experimental "movie" provided as a supplemental file is indeed enjoyable. The complex spatiotemporal propagation appeared in the movie is a clear manifestation of quantum interferences in the molecular rotational WP created therein. The observation was done with a conventional VMI setup, and the detailed and clear signature thus observed is mainly due to the two following efforts by the authors. First, the molecular ensemble to be examined was initially prepared as that restricted in the (almost) single lowest rotational level after the quantum-state selection with the electrostatic deflector, which was originally developed by the authors' research group. They have implemented such state selection of polar molecules to various researches already. Here, the restriction of the initial ensemble to the single level is essential to observe the quantum interference, which would be smeared out if the ensemble were composed of multiple rotational levels. The previous study by our research group (given in reference [11]) was along this direction, and the present report has reached to the limit. The second effort is the achievement of the excitation that makes the WP containing many rotational eigenstates with J ranging from 0 to 15. Such a high degree of state "mixing" inherently results in the rich and complex dynamics appeared in the "movie".

We thank the referee for providing this appropriate summary of our work.

We wish to point out that, in addition, the *flat phase* of the wavepacket at the revivals is crucial for the strong modulations and the high degree of alignment observed in the movie.

2) Since anisotropic molecular ensembles have been useful in various researches in the fields of molecular physics and quantum optics, achieving higher degree of molecular alignment is important. The present study reports the realization of field free molecular alignment exceeding the estimated limit for single-pulse excitation. It is really nice. The author adopted a double-pulse excitation scheme, which has been repeatedly examined and approved for improving molecular alignment, and found out the optimum condition from the model calculation based on the time-dependent Schrodinger equation (TDSE) calculation. Again, the restriction of the initial ensemble to the lowest single level is essential to achieve the highest alignment realized so far, and the development of the electrostatic deflector by the authors is crucial.

Thank you very much for this very positive evaluation.

3) Retrieval of quantum states is also a big issue in molecular physics and quantum optics, and there have been many researches. For molecular rotational WPs, theoretical proposals and experimental demonstration have already been reported [A. S. Mouritzen and K. Molmer, J. Chem. Phys. 124, 244311 (2006); H. Hasegawa and Y. Ohshima, Phys. Rev. Lett. 101, 053002 (2008)], which the authors would cite.

We have added these two references [12, 13], pointed out by the referee, in the introduction of the main manuscript.

The differences between our work and the two studies mentioned by the referee is that

1. in the work of Mouritzen and co-workers [12], a theoretical method is proposed to retrieve the rotational wavepacket, but no reconstruction of experimental data is given since, as the authors stated, large and time consuming data acquisition is needed.

In our approach no tomography is necessary and comparably small datasets suffice to reconstruct the wavepacket.

2. The experimental method and observables used by Hasegawa and Ohshima [13] are also very different from our work. In their work, the yield of benzene cations resulting from (1+1) REMPI using a tunable frequency-doubled nanosecond dye laser was monitored using a time-of-flight mass spectrometer as a function of the time delay between a pair of femtosecond pump pulses. The procedure that was followed allowed to reconstruct the phases and the populations only when restricting the analysis to a single initial rotational state.

In contrast to this work, we recorded 2D angle-resolved ion-momentum distributions from which 2D angular distributions were extracted and compared to simulations. The availability of the (2D) angular distributions allowed us to extract the different moments of the angular distribution and to compare all of them individually/jointly, yielding significantly more information. Furthermore, in our analysis, the mixed nature of the initial state distribution as well as the effect of focal volume averaging was accounted for.

It is true that a series of 2D images for fragment ions from rotating molecules contains rich information on the rotational WP, but its extraction is not straightforward. The ionization/dissociation process is inherently anisotropic. When the linearly polarized light is used to induce Coulomb explosion, the ionization/dissociation probability is symmetric around the polarization direction but strongly depends on the angle between the polarization and the molecular axis. Furthermore, in the case of polyatomic molecules, like OCS, we have to take into account that the fragment recoil velocity may not direct along the molecular axis due to the ultrafast bending of the molecule during the ionization/dissociation. See, for example, J. H. Sanderson et al., Phys. Rev. A 65, 043403 (2002). Thus, when 2D ion imaging is adopted, as the authors did, where ions distributed in 3D are smashed onto the 2D detector, the observed images are given after convoluting the angular probability for the molecular direction (proportional to the absolute square of the WP) over ionization/dissociation anisotropic probability as well as the angular distribution due to the bending dynamics. There is no description about this situation anywhere in the manuscript, and Eq. 2 in Supplemental Information implies that the authors may simply neglect other factors than that originates from WP. This is totally inappropriate. If the authors did make any consideration over other factors, it should be clearly described so that its validity and limitation are properly evaluated.

We agree with the referee that the measured angular distributions are a convolution between the rotational dynamics, induced by the pump laser pulses, the angle-dependent ionisation probability, and possible effects from non-axial recoil due to the bending of the molecule in the strong laser field. However, we point out that in our experiment the effects of angle-dependent ionisation probability are very small and effects from non-axial recoil are negligible.

We have now included the angle-dependence of the ionization process in our simulations by including in the calculated 3D velocity distributions, from which the 2D momentum distributions are obtained, the square of the measured angle-dependent single-electron ionization probability. The latter is used to account for the double ionisation process that led to the formation of a doubly charged cation that quickly Coulomb explodes to form the O^+ ions that are used in our experiment to quantify the degree of molecular alignment. This procedure has actually resulted in an improvement of the agreement between the angular distributions, decomposed in terms of Legendre polynomials, extracted from the measurement and from the calculation. On page 3 of the main manuscript, we have added the following sentence: “The anisotropic angle-dependent

ionisation efficiency for double ionisation, resulting in a two-body breakup into O^+ and CS^+ fragments, was taken into account by approximating the probe factor through the square of the measured single-electron ionisation rate.” Figure 2 of the main manuscript has also been updated with a new comparison between experiments and calculations.

Regarding the work by Sanderson et al. [14], discussed by the referee in the context of non-axial recoil of ionic fragments of OCS during the fragmentation process, we wish to provide the following comment: The effect of non-axial recoil critically depends on the exact experimental parameters, such as the wavelength of the ionising laser, the pulse duration, the peak intensity and the charge states created during the fragmentation process. Sanderson et al. [14] used a central wavelength of 790 nm and a peak intensity above 10^{15} W/cm² and observed highly charged ionic fragments, created through three-body fragmentation. In our experiment, the peak intensity was more than one order of magnitude lower whilst using a wavelength of 1.75 μ m, both of which lead to a strongly reduced ionisation probability. This is reflected in our experiment, where singly charged oxygen ions, created through a two-body break up with a singly charged CS^+ counterion, were used as observable. Therefore, a direct transfer of the results of [14] to our experiment is not possible. Moreover, the effect of bending of the molecule during Coulomb explosion, i. e., non-axial recoil, would lead to a broadening of the angular distribution for an aligned molecular ensemble and hence to a lower experimental value of $\langle \cos^2 \theta_{2D} \rangle$. In that sense, our measurements constitute a lower bound of the actual degree of alignment, which we stated in the main manuscript on page 2.

On the basis of the above arguments we believe that neglecting non-axial recoil during the fragmentation process is well justified in our analysis. We have now added on page 3 the following sentence: “Non-axial recoil during the fragmentation process is expected to be negligible and was not included in the simulations.”

In addition to the above main points, there are several comments to be considered.

i) In FIG.3. $\phi(J)$ is used without definition. Is it a total phase: $\phi(J) = \omega_J t + \delta_J$?

The $\phi(J, t)$ are the phases of the complex coefficients $a_J(t)$ of the wavepacket, created by the alignment laser pulses. In Figure 3 of the main manuscript, these phases are plotted for the initial state $|J = 0, M = 0\rangle$, which corresponds to the initial state with the largest population; see the initial state distribution in the supplementary information. As mentioned by the referee, this phase can be written as $\phi(J) = \omega_J t + \delta_J$, with δ_J being the phase accumulated during the interaction with the laser pulse. We have added, on page 3 of the main manuscript, the following sentence: “In Figure 3b the corresponding phase differences for all populated states relative to the initial state with the largest population in the wavepacket are shown, where $\phi(J)$ is the phase of the complex coefficient a_J in (1).”

ii) In Supplemental Information, p.4. It has to be clearly described: what are the floating parameters in the least squares fittings? Amplitudes and phases for each eigen functions (as has to be in conventional WP reconstruction, see Hasegawa and Ohshima), or others (e.g., laser pulse parameters)?

The rotational wavepacket for each initially populated state $|J_i, M_i\rangle$, was fully calculated by solving the time-dependent Schrödinger equation for the case of a rigid rotor. The laser parameters, such as the energies of the two pulses and their common duration, were fixed to the experimentally determined values. The initial state distribution, together with the radial waist of both, the pump and the probe laser beams that were used in the focal averaging, were used as fitting parameters. We have expanded and clarified the second section of the supplementary information, now providing a more detailed description of the fitting procedure and the floating parameters in the least squares fitting.

iii) P. 4, in the main text. What is the difference between the “2D-” and “3D-” reconstruction? What is the experimental input for the reconstruction (to fit the parameters) and what are the

fitting parameters, for the both cases?

For the complete reconstruction of the wavepacket, the initial conditions for solving the TDSE are needed. We obtained the initial conditions consisting of the initial state distribution and the laser focal sizes by fitting the simulated 8 lowest-order even moments to the experimental ones, for different values of the initial states populations and laser foci, until best agreement was achieved, characterised by least squares minimisation. All other experimental parameters, described on page 6 of the supplementary information, were known or estimated from the experiment. Having all information needed for solving the TDSE for all initial states, we obtain the final 3D time-dependent rotational density as an incoherent average over all wavepackets, weighted by their initial population – as obtained from the fitting procedure. In order to compare the simulations directly to our measurements, we used a Monte-Carlo routine to project the simulated 3D rotational density onto a 2D screen. A direct Abel-inversion to retrieve the experimental 3D distributions was not possible due to the polarization of the ionising laser pulse, which was chosen to be perpendicular to the detector plane to avoid any influence from geometric alignment, *vide supra*. In this sense, there is no 2D and 3D reconstruction, the reconstruction allows to retrieve the 3D rotational density, which is then projected onto a 2D screen for comparison to the experiment. We have included a figure in the supplementary information, Figure S2, that schematically shows the 3D rotational density and the 2D projected image, explaining the relation between the angles θ and θ_{2D} .

iv) In Supplemental Information, p.6. In an analysis (3D?), the initial state distribution with $J = 0$ to 2 is considered. How about the actual population for each state?

The initial state distribution w_{JM} as extracted from the fitting procedure and used in the incoherent average – see eq. 4 of supplemental material – are

$$\begin{array}{lll} w_{00} = 0.82 & & \\ w_{10} = 0.037 & w_{11} = 0.075 & \\ w_{20} = 0.015 & w_{21} = 0.021 & w_{22} = 0.032 \end{array}$$

We included the initial state populations in the supplementary information in section 'Moments of angular distribution' on page 6.

v) It should be pointed out, when the initial state distribution is spread among multiple levels, the resultant molecular ensemble after laser excitation cannot be represented as a single WP, but a mixed state. Even when the initial ensemble is in a single state, if the laser field variation is substantial in the probed area, the resultant ensemble is also in a mixed state. For both cases, a density matrix should be invoked to describe the state and any "WP reconstruction" cannot be adopted.

Indeed, the most complete approach is to utilize the density matrix, when working with ensembles of molecules being distributed over many states and averaging over the focal volume. We took this explicitly into account by averaging over the initial state distribution in eq. 4 and the focal volume in eq. 5 of the supplement. The fitting parameters were the populations of these initial states and the laser focal parameter. The wavepackets, emanating from each initial state and for each intensity in the focus, are incoherently summed. Thus, our approach – as described in the supplementary material – is equivalent to using a density matrix. We have adapted the description of the fitting procedure and the projection of the 3D rotational density to 2D, which allowed direct comparison with the experiment, in the supplementary information.

In conclusion, the manuscript contains very interesting finding pertinent to coherent rotational dynamics of an ultrashort-laser driven molecular ensemble. Still, there are several points, in particular mentioned in 3) above, to be seriously considered and substantially revised before publication.

We thank the referee for his interest in our work and the strong support. We hope that we could satisfactorily respond to all his questions and comments, which helped to improve the quality of the manuscript. Thus, we look forward to the publication of the improved manuscript in Nature Communications.

Remarks by referee 2

This manuscript presents a superb work by a team, which includes some of the pioneers of molecular alignment studies. The work utilizes all advanced alignment and detection techniques to achieve the highest degree of field-free alignment of OCS and to record a very detailed high-resolution images of time-dependent rotational nuclear wavepackets. I could not find any faults with either experiments or calculations. The paper is very well written, the results are clearly presented, the methods and approaches are well documented (in the paper and supplementary materials). The movie also looks cool.

We thank the referee for this very positive evaluation of our work and his confirmation that our work is absolutely sound, correct, and that the presentation is even “cool”.

However, I cannot recommend this work for publication in Nature Communications. My opinion is that this work does not constitute a genuine breakthrough. While the authors seem to consider and do emphasize as their main achievement the high-resolution “molecular movie” of nuclear rotation, that process is generic for all linear molecules, very easily modelled numerically with an arbitrarily high accuracy and has been observed many times before, including by the authors of this manuscript themselves. There is very little actual benefit in the demonstrated high angular resolution in terms of gaining new insights or improving numerical models. Therefore, I am not so very impressed with the “movie” part.

We point out that we are not aware of any observation of these effects with even nearly the same fidelity and dynamics. Furthermore, as pointed out at the beginning of this reply letter, molecular alignment and the knowledge of the complete rotational wavepacket is of great importance for many experiments. For instance, in angularly resolving imaging experiments the angular differential cross section will be convoluted by the angular distribution of the molecules. Here, the achieved $\langle \cos^2\theta \rangle = 0.94$ provides still good angular contrast, whereas any alignment $\langle \cos^2\theta \rangle < 0.9$ would compromise the angular resolution in an unacceptable manner [15]. Thus, the very strong degree of alignment presented by us provides the necessary angular control for such imaging experiments, including molecular-frame laser-induced electron diffraction.

Although several methods to retrieve rotational wavepackets were theoretically proposed [7, 16, 17], up to now most work on aligned molecules considered only the second moment $\sim \langle \cos^2\theta \rangle$ of the angular distribution to characterise its degree of anisotropy. Only in few experiments were higher-order moments of the angular distribution discussed [18]. Thus, the work we present here is unique in its quality and the wavepacket reconstruction is the most accurate to date.

What I am impressed with (and what, I think, is the real main achievement of this work) is how it combines all the advanced techniques: rotational state selection, two-pulse impulsive alignment, Coulomb explosion imaging, VMI - to achieve near-perfect field-free alignment.

We thank the referee for this again very positive judgement of our work. We wish to point out that it is indeed the unique combination of all these highly advanced techniques that enabled the presented record-high degree of alignment, but also the strongly modulated dynamics and the observation thereof.

Having said that, I will also note that all those techniques were already demonstrated separately some time ago, also by the authors themselves, and those papers are properly cited here too.

Combining the established techniques is rather an incremental development, which while novel and important, in my view does not rise to the standard of Nature Communications. I appreciate, that this is a subjective judgement call based on comparing this work with the whole output of the journal, which is, perhaps, best done by the editors. Also, those dreaded "novelty and importance" are not so easily quantified, so I have to go purely with my subjective opinion here. I feel that this manuscript would be more appropriate for Scientific Reports.

We feel that the elaborations in this letter and the improved manuscript provide also clear evidence for the importance of our work.

Remarks by referee 3

In this manuscript the authors describe alignment experiments where a pair of nonresonant pulses generates a broad rotational wavepacket and Coulomb imaging is applied to probe the rotational distribution as a function of time. It is a beautiful paper and I recommend accepting it for publication in Nature Communications. I have, however, several comments that the authors may wish to consider.

We thank the referee for this very positive evaluation and the support of our work.

1. The authors should explain the difference between $\langle \cos^2\theta_{2D} \rangle$ and $\langle \cos^2\theta \rangle$.

In our manuscript, θ is defined as the Euler angle that defines the orientation/alignment of the molecular axis with respect to the pump laser polarization axis, whereas θ_{2D} corresponds to the angle between the laser polarization axis and the detected ion momentum distribution on the 2D detector plane. The relationship between the two quantities is actually not straightforward. As a matter of fact, in our calculation, we calculated the angular distribution of the molecules in the laboratory frame, i. e., $\langle \cos^2\theta \rangle$. The comparison with the experimental observation was done through a projection, using a Monte Carlo sampling routine, of the 3D alignment distribution onto a 2D plane, mimicking the detector plane, in order to extract, from the calculations, $\langle \cos^2\theta_{2D} \rangle$ and the higher-order Legendre polynomials. The relation between the rotational density in 3D and the 2D projected images is now schematically illustrated in the new Figure S2 in the supplementary information, also sketching the two angles θ and θ_{2D} .

2. What is the accuracy of the measurement? Is the relative error small with respect to the difference between 0.96 and 0.92 (where the latter figure corresponds to a single initial eigenstate and one pulse)? I expect that the experimental accuracy is limited by the accuracy to which the intensity can be measured. With the error taken into account, 0.96 is not very different from the single pulse 0.92. This suggests that the good alignment demonstrated here is predominantly because of the single initial eigenstate. This agrees with early calculations.

We optimized and measured also single pulse alignment, obtaining a maximum degree of field-free alignment of $\langle \cos^2\theta_{2D} \rangle = 0.92$. The angular distribution in this case is clearly different from $\langle \cos^2\theta_{2D} \rangle = 0.96$ with an opening angle (FWHM) of 17.5° , compared to 13.4° for the two-pulse experiment. A plot with a comparison of the two angular distributions, extracted from the experiment, is shown in Figure 1 (of this letter). The maximum degree of alignment achieved with a single pulse was extracted at the half revival at a delay time of 42.75 ps after the alignment laser pulse, having a peak intensity of $I_{\text{align}} = 8 \cdot 10^{12} \text{ W/cm}^2$. In the case of two-pulse alignment, the maximum degree of alignment was extracted at the 3/2 revival at a delay time of 120.78 ps after the first alignment pulse with the first pulse having a peak intensity of $I_{\text{align},1} = 1.9 \cdot 10^{12} \text{ W/cm}^2$ and the second pulse $I_{\text{align},2} = 5.5 \cdot 10^{12} \text{ W/cm}^2$. The accuracy to which the degree of alignment can be characterised is not related to the laser intensity since the value of $\langle \cos^2\theta_{2D} \rangle = 0.96$ was directly extracted from the measured 2D momentum maps and so no intensity fitting was necessary. We agree that the initial state distribution plays a major role

Figure 1: Comparison of angular distributions for optimized single and two-pulse alignment, extracted from experiment. The angular distributions are clearly different with opening angles of 17.5° and 13.4° , respectively. Both angular distributions were normalized to unit area.

for the alignment dynamics and especially for the degree of alignment that can be reached, with a degree of alignment that decreases with an increasing number of initially populated states.

Figure 1 clearly shows that we are sensitive to such “small” changes of the $\langle \cos^2 \theta_{2D} \rangle$ value. Moreover, our sensitivity to differences on that order were also confirmed by many computational results for finite temperatures and different laser-pulse parameters.

3. Equation (1) and the details below are not needed. The dynamics of rotational wavepackets of simple, rigid rotor molecules have been computed before and the method is known. The authors could provide a reference to a theory paper or review where this is more fully explained and use the space to expand on the results.

We believe that explicitly stating equation (1) is important, especially in view of the broad audience of Nature Communications. Since this equation is at the heart of observed and disentangled wavepacket dynamics, we believe that it is crucial to state this equation for the understanding of the work by non-experts.

4. P. 2, paragraph 2, 6 lines from the end: Why do the authors use Chebyshev polynomials? Please see Phys.Rev.Lett. 89 233002 (2002), from which it is evident that the correct expansion polynomials are the Legendre polynomials (or the Legendre functions in the more general case). The cross section could be expanded in terms of other polynomials but this does not have physical origin. Expansion in terms of Legendre polynomials yields the moments of the rotational distribution as the expansion coefficients. The same reference makes the argument made also here that the completer rotational density contains much more information (all moments) than the conventional $\langle \cos^2 \theta \rangle$.

We agree with the statement that the natural expansion is in terms of Legendre polynomials, and we have updated the manuscript accordingly, i. e., we changed Figure 2 of the main manuscript and Figure 3 of the supplementary information accordingly.

We note that from a mathematical point of view either choice is totally valid and the two expansions are completely equivalent. We have now carried out our analysis in terms of both, Legendre and squared Chebyshev polynomials, and obtained the same agreement between experiment and simulation, as stated in the supplementary material. Furthermore, we want to point out that the expansion of the “total” angular distribution in 3D in terms of Legendre polynomials is connected to the cross section, but not the 2D expansion we carry out in this work.

5. Figure 2b: The authors argue several times that the agreement of the calculations with the measurements is excellent, but in reality, considering that the calculation used radial distributions extracted from the experiments, one could expect much better agreement of the images. Why is the resemblance poor?

We have improved the simulated images, as can be seen in the new movie and the new Figure 2 in the main manuscript, by now including the experimental radial distribution in 3D, prior to the projection onto the detector, which yields much better visual agreement. In terms of the physics and the derived parameters, nothing has changed and the results are identical to before.

6. The experimental images are not a measurement of the square magnitude of the wavepacket (as the manuscript argues) because they include the complex and system-dependent dynamics of ionization and fragmentation. Of course, the rotational distribution has a role in determining the images, basically because the measured angular distribution is tied to the rotational wavepacket angular distribution through angular momentum selection rules. Please see *Phys.Rev. A* **87**, 023411 (2013) for explanation how the rotational wavepacket angular density relates to the observed image.

We thank the referee for this important comment. We are aware of this issue and the reference. We did not include fragmentation dynamics in our calculations, which would be far beyond the scope of this work. Following one of the comments of the first referee, we now included the angle-dependent ionisation probability in our simulations, using measured ionisation yields as a function of the angle between alignment and probe laser. However, we did not include effects, such as non-axial recoil in the fragmentation process, which to our knowledge is negligible for the intensity used in our experiment, *vide supra*.

We thank the referees for their constructive feedback! We believe that we have addressed all comments of all referees in detail and feel that it made the paper even stronger. Based on the points given above, we consider this manuscript of broad and high importance and hope that it can be accepted and published in Nature Communications in due course.

Along the resubmitted manuscript and supplementary information text we also provide PDFs with all changes clearly marked.

References

- [1] F. Rosca-Pruna and M. J. J. Vrakking, “Experimental observation of revival structures in picosecond laser-induced alignment of I_2 ,” *Phys. Rev. Lett.* **87**, 153902 (2001).
- [2] J. J. Larsen, I. Wendt-Larsen, and H. Stapelfeldt, “Controlling the branching ratio of photodissociation using aligned molecules,” *Phys. Rev. Lett.* **83**, 1123–1126 (1999).
- [3] J. J. Larsen, K. Hald, N. Bjerre, H. Stapelfeldt, and T. Seideman, “Three dimensional alignment of molecules using elliptically polarized laser fields,” *Phys. Rev. Lett.* **85**, 2470–2473 (2000).
- [4] J. Itatani, J. Levesque, D. Zeidler, H. Niikura, H. Pépin, J. C. Kieffer, P. B. Corkum, and D. M. Villeneuve, “Tomographic imaging of molecular orbitals,” *Nature* **432**, 867–871 (2004).
- [5] M. Meckel, D. Comtois, D. Zeidler, A. Staudte, D. Pavičić, H. C. Bandulet, H. Pépin, J. C. Kieffer, R. Dörner, D. M. Villeneuve, and P. B. Corkum, “Laser-induced electron tunneling and diffraction,” *Science* **320**, 1478–1482 (2008).

- [6] L. Holmegaard, J. L. Hansen, L. Kalhøj, S. L. Kragh, H. Stapelfeldt, F. Filsinger, J. Küpper, G. Meijer, D. Dimitrovski, M. Abu-samha, C. P. J. Martiny, and L. B. Madsen, “Photoelectron angular distributions from strong-field ionization of oriented molecules,” Nat. Phys. **6**, 428 (2010), arXiv:1003.4634 [physics] .
- [7] C. Marceau, V. Makhija, D. Platzer, A. Y. Naumov, P. B. Corkum, A. Stolow, D. M. Villeneuve, and P. Hockett, “Molecular frame reconstruction using time-domain photoionization interferometry,” Phys. Rev. Lett. **119**, 083401 (2017).
- [8] J. Küpper, S. Stern, L. Holmegaard, F. Filsinger, A. Rouzée, A. Rudenko, P. Johnsson, A. V. Martin, M. Adolph, A. Aquila, S. Bajt, A. Barty, C. Bostedt, J. Bozek, C. Caleman, R. Coffee, N. Coppola, T. Delmas, S. Epp, B. Erk, L. Foucar, T. Gorkhover, L. Gumprecht, A. Hartmann, R. Hartmann, G. Hauser, P. Holl, A. Hömke, N. Kimmel, F. Krasniqi, K.-U. Kühnel, J. Maurer, M. Messerschmidt, R. Moshhammer, C. Reich, B. Rudek, R. Santra, I. Schlichting, C. Schmidt, S. Schorb, J. Schulz, H. Soltau, J. C. H. Spence, D. Starodub, L. Strüder, J. Thøgersen, M. J. J. Vrakking, G. Weidenspointner, T. A. White, C. Wunderer, G. Meijer, J. Ullrich, H. Stapelfeldt, D. Rolles, and H. N. Chapman, “X-ray diffraction from isolated and strongly aligned gas-phase molecules with a free-electron laser,” Phys. Rev. Lett. **112**, 083002 (2014), arXiv:1307.4577 [physics] .
- [9] C. J. Hensley, J. Yang, and M. Centurion, “Imaging of isolated molecules with ultrafast electron pulses,” Phys. Rev. Lett. **109**, 133202 (2012).
- [10] J. Yang, M. Guehr, T. Vecchione, M. S. Robinson, R. Li, N. Hartmann, X. Shen, R. Coffee, J. Corbett, A. Fry, K. Gaffney, T. Gorkhover, C. Hast, K. Jobe, I. Makasyuk, A. Reid, J. Robinson, S. Vetter, F. Wang, S. Weathersby, C. Yoneda, M. Centurion, and X. Wang, “Diffractive imaging of a rotational wavepacket in nitrogen molecules with femtosecond megaelectronvolt electron pulses,” Nat. Commun. **7**, 11232 (2016).
- [11] M. G. Pullen, B. Wolter, A.-T. Le, M. Baudisch, M. Hemmer, A. Senfleben, C. D. Schroter, J. Ullrich, R. Moshhammer, C. D. Lin, and J. Biegert, “Imaging an aligned polyatomic molecule with laser-induced electron diffraction,” Nat. Commun. **6**, 7262 (2015).
- [12] A. S. Mouritzen and K. Mølmer, “Quantum state tomography of molecular rotation,” J. Chem. Phys. **124**, 244311 (2006).
- [13] H. Hasegawa and Y. Ohshima, “Quantum state reconstruction of a rotational wave packet created by a nonresonant intense femtosecond laser field,” Phys. Rev. Lett. **101**, 053002 (2008).
- [14] J. H. Sanderson, T. R. J. Goodworth, A. El-Zein, W. A. Bryan, W. R. Newell, A. J. Langley, and P. F. Taday, “Coulombic and pre-coulombic geometry evolution of carbonyl sulfide in an intense femtosecond laser pulse, determined by momentum imaging,” Phys. Rev. A **65**, 043403 (2002).
- [15] F. Filsinger, G. Meijer, H. Stapelfeldt, H. Chapman, and J. Küpper, “State- and conformer-selected beams of aligned and oriented molecules for ultrafast diffraction studies,” Phys. Chem. Chem. Phys. **13**, 2076–2087 (2011), arXiv:1009.0871 [physics] .
- [16] X. Wang, A.-T. Le, Z. Zhou, H. Wei, and C. D. Lin, “Theory of retrieving orientation-resolved molecular information using time-domain rotational coherence spectroscopy,” Phys. Rev. A **96**, 023424 (2017).
- [17] S. Ramakrishna and T. Seideman, “Rotational wave-packet imaging of molecules,” Phys. Rev. A **87**, 023411 (2013).

- [18] P. W. Dooley, I. V. Litvinyuk, K. F. Lee, D. M. Rayner, M. Spanner, D. M. Villeneuve, and P. B. Corkum, “Direct imaging of rotational wave-packet dynamics of diatomic molecules,” Phys. Rev. A **68**, 023406 (2003).

Reviewers' comments:

Reviewer #1 (Remarks to the Author):

The revised manuscript seems to clarify most of the inquiries I have raised. In particular, consideration of radial distribution for two-body Coulomb explosion has yielded much improved close-up between the observed and calculated "movies," which is really nice. So I think this manuscript can be worth published in Nature Communication, after some consideration about the following points.

1) In the main text, authors used several times "reconstruction of a rotational wavepacket" or similar notation for describing an achievement of the present study. I should point out, in the researches in quantum optics and molecular physics, a term "quantum-state reconstruction" is usually used as a procedure for determining or characterizing a quantum state in which detailed prior knowledge of the Hamiltonian is not required. See, for example, a classical paper on this subject [T. S. Humble and J. A. Cina, Phys. Rev. Lett. 93, 060402 (2004)]. On the other hand, the authors actually have determined some parameters of the ultrashort pulses to create the wavepackets and initial rotational-state distribution to reproduce the observed angular probability distribution. Readers of the manuscript may have some confusion due to the difference in the definitions. It is recommended that the descriptions relating to "reconstruction" will be appropriately revised.

2) In page 4 of the main text, line 9 of right column. A word "initial" is unnecessary?

3) In page 6 of the Supplementary Information. It should be recommended to indicate statistical errors for the parameters determined by the least-squares fitting.

4) In page 8 of the Supplementary Information. The similar observation on different angular distributions for the same degree of alignment has already reported in Ref. [12] of the main text, even though the quality of the data has now been much improved.

Remarks by reviewer 1

The revised manuscript seems to clarify most of the inquiries I have raised. In particular, consideration of radial distribution for two-body Coulomb explosion has yielded much improved close-up between the observed and calculated “movies,” which is really nice. So I think this manuscript can be worth published in Nature Communication, after some consideration about the following points.

1) In the main text, authors used several times “reconstruction of a rotational wavepacket” or similar notation for describing an achievement of the present study. I should point out, in the researches in quantum optics and molecular physics, a term “quantum-state reconstruction” is usually used as a procedure for determining or characterizing a quantum state in which detailed prior knowledge of the Hamiltonian is not required. See, for example, a classical paper on this subject [T. S. Humble and J. A. Cina, Phys. Rev. Lett. 93, 060402 (2004)]. On the other hand, the authors actually have determined some parameters of the ultrashort pulses to create the wavepackets and initial rotational-state distribution to reproduce the observed angular probability distribution. Readers of the manuscript may have some confusion due to the difference in the definitions. It is recommended that the descriptions relating to “reconstruction” will be appropriately revised.

We understand that there might be some confusion with respect to the technicalities of terms like “quantum-state reconstruction” or “quantum-state tomography” and, therefore, we have adjusted our wording accordingly throughout the manuscript and the SI. The exact changes should be obvious from the PDF with changes marked in colour. We have not replaced the term in the introduction of references 21 and 22, as here this term is cited from these references.

2) In page 4 of the main text, line 9 of right column. A word “initial” is unnecessary?

Indeed – thank you for pointing this out. We have removed the ‘initial’.

3) In page 6 of the Supplementary Information. It should be recommended to indicate statistical errors for the parameters determined by the least-squares fitting.

We have added this information to the supplementary information.

4) In page 8 of the Supplementary Information. The similar observation on different angular distributions for the same degree of alignment has already reported in Ref. [12] of the main text, even though the quality of the data has now been much improved.

We have added a reference to Fig. 10 of this paper to the caption of Fig. S4 in our supplementary information.

REVIEWERS' COMMENTS:

Reviewer #1 (Remarks to the Author):

The revised manuscript seems to clarify all the inquiries raised in the previous review. So I think this manuscript can now be worth published in Nature Communication.